

# Variational Bayesian phylogenies through matrix representation of tree space

Remco R. Bouckaert

School of Computer Science, University of Auckland, Auckland, New Zealand

## ABSTRACT

In this article, we study the distance matrix as a representation of a phylogeny by way of hierarchical clustering. By defining a multivariate normal distribution on (a subset of) the entries in a matrix, this allows us to represent a distribution over rooted time trees. Here, we demonstrate tree distributions can be represented accurately this way for a number of published tree distributions. Though such a representation does not map to unique trees, restriction to a subspace, in particular one we call a "cube", makes the representation bijective at the cost of not being able to represent all possible trees. We introduce an algorithm "cubeVB" specifically for cubes and show through well calibrated simulation study that it is possible to recover parameters of interest like tree height and length. Although a cube cannot represent all of tree space, it is a great improvement over a single summary tree, and it opens up exciting new opportunities for scaling up Bayesian phylogenetic inference. We also demonstrate how to use a matrix representation of a tree distribution to get better summary trees than commonly used maximum clade credibility trees. An open source implementation of the cubeVB algorithm is available from https://github.com/rbouckaert/cubevb as the cubevb package for BEAST 2.

## INTRODUCTION

The workhorse of Bayesian phylogenetics is the Markov chain Monte Carlo (MCMC) algorithm, which is widely used to reconstruct the evolutionary relationships among a set of biological sequences (*Huelsenbeck & Ronquist, 2001*; *Höhna et al., 2016*; *Suchard et al., 2018*; *Bouckaert et al., 2019*). In practice, MCMC is limited to handling phylogenies of at most a few thousand taxa. With the advent of ever cheaper sequencing techniques, these algorithms are overwhelmed by the amount of available data. Consequently, maximum likelihood methods are employed, which can handle these amounts of data because they are based on optimisation algorithms.

Variational Bayesian (VB) methods (*Jordan et al., 1999*) are also based on optimisation, so offer the prospect of being more scalable than MCMC. VB methods usually represent a posterior distribution as a transformation of a multivariate normal distribution on Euclidean space. The form of the transformation is crucial in how accurate the posterior distribution can be represented. Usually, stochastic gradient ascent is used as the optimisation algorithm, which relies on the transformation being a bijection between trees and their transformed representation. This provides a major hurdle for

Corresponding author
Remco R. Bouckaert,
remco@cs.auckland.ac.nz

phylogentic inference, since existing tree spaces are either not bijective (like spanning trees between points in hyperbolic space (*Matsumoto, Mimori & Fukunaga, 2021*; *Jiang, Tabaghi & Mirarab, 2022*)) or do not allow representation as a Euclidean space (like ultrametric space (*Gavryushkin & Drummond, 2016*)).

Prior phylogenetic VB implementations got around these restrictions by being limited to fixed tree topologies (*Fourment & Darling, 2019*; *Zhang, 2020*). Others do not handle timed trees (*Zhang & Matsen IV, 2018*) which are important to many practical and scientific questions. The ones that handle time trees (*Zhang & Matsen IV, 2022*), have not demonstrated to scale to scale to 1,000 taxa.

Here, we explore matrix representations of tree space. The basic idea is to use a symmetric matrix with dimension equal to the number of taxa and apply a hierarchical clustering algorithm like the single link clustering algorithm (*Florek et al., 1951*) to obtain a tree with internal node heights specified by the values of the matrix. The entries in the matrix form a Euclidean space, however there are many ways to represent the same tree, so the transformation is not bijective. A ''cube'' is a matrix where we restrict ourselves to the 1-off-diagonal entries of the matrix (and leave the rest at infinity) and the transformation becomes a bijection, but cannot represent all possible trees any more. In this article, we demonstrate it is possible to capture the most interesting part of posterior tree space using this approach. We implement an algorithm for basic phylogenetic models, infer the parameters of the multivariate normal distribution and show through a well calibrated simulation study that it captures important parameters of interest. We demonstrate that it scales well.

Representations similar to the cube representation have been used before in phylogenetic inference. For example, in training neural nets to uncover the epidemiological dynamics of outbreaks (*Voznica et al., 2022*), where the ''compact bijective ladderised vector'' orders taxa so that branch support for taxa on the left is maximised. Instead of using the distance between taxa, the distance from internal nodes to the root is used. In MCMC, tree proposals like the ''node reheight'' operator in *BEAST (*Heled & Drummond, 2009*), order taxa through a tree traversal. At binary splits, the first or second branch is randomly traversed first, resulting in a random ordering. Then, an internal node is randomly selected, and its height is randomly moved (constrained by gene tree heights). Finally, a tree is reconstructed to maintain consistency with the ordering and node heights of internal nodes between taxa. *Mau, Newton & Large (1999)* refers to this as the ''canonical representation'' and proposes its use an MCMC proposal as well. This is equivalent to selecting a cube with random order consistent with the tree, then perturbing a distance in the matrix and reconstructing the tree.

Matrix and specifically cube representations offer a few other benefits. Many post-hoc analyses such as ancestral reconstruction, trait inference or phylogeography are performed on a single summary tree, so these analyses do not take phylogenetic uncertainty in account. Though a cube does not represent all of tree space, in many situations it captures most of the uncertainty of a posterior distribution, and allows taking uncertainty in account in post-hoc analyses. Furthermore, our novel way to represent tree space opens up new avenues to implement online algorithms (*Bouckaert, Collienne & Gavryushkin, 2022*), allowing rapid
responses to newly available data. There is also potential for improving summary trees through matrix representation of a given tree set, as we will demonstrate.

## METHODS

### Representing trees through a matrix

The main idea is to represent a posterior tree distribution as a distribution over entries of a matrix, and apply the single links clustering algorithm (Algorithm 1) to obtain a tree. The matrix can be interpreted as a distance matrix: it must be symmetric, have positive values and the entries on the diagonal are ignored. So, for a tree with $n$ taxa there are $n(n-1)/2$ entries that are relevant: the entries above the diagonal. We say a matrix entry is *specified* if it is finite, and *unspecified* if it is positive infinite. We say a matrix is *valid* if the tree associated with it through single link clustering only has finite branch lengths.

---

**Algorithm 1** Single link clustering

Input: $n \times n$ matrix M with entries $m_{ij}$
Let $x_1, \ldots, x_n$ be $n$ taxa
$\forall_{1 \leq i \leq n}$ initialise subtree as single taxon $\{x_i\}$ at height 0
**while** There is more than 1 subtree **do**
    Let $m_{ij}$ be minimum of $M$, where $x_i$ and $x_j$ in different subtrees
    Create node $x$ with height $m_{ij}$
    Add branch between $x$ and root of subtree containing $x_i$
    Add branch between $x$ and root of subtree containing $x_j$
Return tree with root at last added node

---

Some properties of this representation with a short justification:

1. *Note that at least $n-1$ entries must be specified for a matrix to be valid.* This follows from the fact that there are $n-1$ internal nodes in a tree, so the single link clustering algorithm requires at least $n-1$ finite heights to specify a tree.

2. *An adjacency graph $G$ over $n$ nodes $v_1, \ldots, v_n$ can be associated with a matrix $M$ having edges $e_{ij}$ between $v_i$ and $v_j$ where ever $m_{ij}$ is specified. A matrix is valid if and only if the associated adjacency graph is connected.* This follows from the fact the single link clustering algorithm uses entries in the matrix associated with edges in the graph that form a path connecting all nodes. So, it is possible for a matrix to have $n-1$ entries specified, but still not be valid (Fig. 1).

3. *To represent a distribution over all possible trees, all entries in the matrix must be specified.* This follows from the observation that in the set of all trees, for every pair $i, j, i \neq j$ of taxa $x_i$ and $x_j$ there exists a tree somewhere in the set of all trees such that the pair is connected with the lowest internal node (that is parent to both taxa). Since such a *cherry* can only be formed by the distance in the matrix $m_{ij}$ being the smallest, this implies $m_{ij}$ cannot be infinite, so $m_{ij}$ must be specified.

4. *A matrix with n or more entries specified does not form a bijection to trees.* Since the single link algorithm only uses $n-1$ entries in the matrix to form the tree, if there

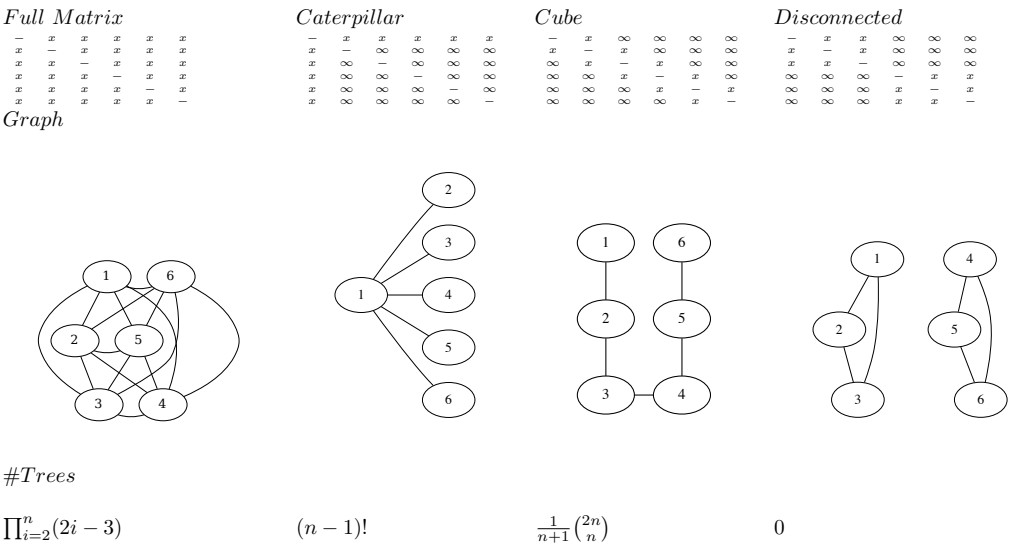

**Figure 1** Various matrices where $x$ means the entry is specified, and $\infty$ the entry is unspecified for $n = 6$, the associated adjacency graphs for $n = 6$ and the number of tree topologies they can represent (in decreasing order).

are $n$ or more values specified in the matrix it is possible to change these values and return the same tree. Therefore, there are multiple ways to represent the same tree with different matrices, hence it cannot be a bijection.

5. *A matrix with $n - 1$ entries specified forms a bijection to trees when the associated adjacency graph is connected.* This includes the cube representation that we will have a closer look at later.

6. *Transforming a tree to a matrix consists of populating the specified entries $m_{ij}$ with height of the most common recent ancestor of $x_i$ and $x_j$ in the tree.* The single link clustering algorithm then retrieves the original tree when applied to the matrix.

If all entries in the matrix on the first row are specified (*i.e.* entries $m_{1j}$ for $1 < j \le n$) and the remainder are not specified (Fig. 1), all possible caterpillar trees with cherry containing $x_1$ can be formed. Note that there are $(n - 1)!$ topologies that can be formed that way.

If all entries in the matrix that are 1-off-diagonal are specified, the adjacency graph forms a linked chain and $\frac{1}{n+1}\binom{2n}{n}$ (the Catalan number) number of topologies can be represented. We call this configuration a *cube*. Even though the same number of entries are specified, this is less than for the caterpillar trees, but will turn out to be more useful in practice. Note that by reordering the taxa it is possible to get the same kind of adjacency graph containing all nodes in a chain.

We will use a multivariate normal distribution $N(\mu, \Sigma)$ with mean vector $\mu$ and covariance matrix $\Sigma$ of dimension $k$. Here $n - 1 \le k \le n(n - 1)/2$, where each of the $k$ dimensions is associated with an entry in the matrix. To estimate $\mu$ and $\Sigma$ for a tree set, simply construct the branch length matrix for each tree in the set and take the average for each entry (that is specified) and covariance for each pairs of specified entries.

To get a sample of trees, draw a random sample from this distribution, exponentiate each of the values and populate the matrix with its values. Then run single link clustering to obtain the tree. Algorithm 2 shows this in more detail.

---

**Algorithm 2** Sample from posterior

---

Input: Multivariate normal distribution $N(\mu, \Sigma)$ of dimension $k$
    where $\mu$ the vector of mean values and $\Sigma$ the covariance matrix.
Input: list of $k$ matrix entries
Let $\mathbf{A}\mathbf{A}^T = \Sigma$ be the Cholesky decomposition of $\Sigma$
Draw $k$ independent standard normal values $\mathbf{z} = \{z_1, \ldots, z_k\}$
Let $\mathbf{x} = \mu + \mathbf{A}\mathbf{z}$
Fill the matrix $M$ with the $k$ entries of $exp(\mathbf{x})$ and leave the rest unspecified
Return tree by applying single link clustering on $M$

---

## Cube space

An alternative way to represent a cube configuration in a matrix is to consider an order of the taxa and a vector that represents the height of the internal nodes between two consecutive taxa in the ordering. Figure 2 illustrates this: the height $h_1$ corresponds to a matrix entry for $A$ and $B$, the height $h_2$ for $B$ and $C$ *etc.*, this way forming the cube configuration from the previous subsection. But we can consider this an ordering $A, B, C, D, E$ and gaps $h_1$ to $h_4$ between consecutive nodes in the ordering.

One nice property of cube space is that it only contains $n - 1$ heights, and forms a bijection under the single link clustering transformation. However, it is obviously limited in that it cannot represent all of tree space, our expectation is that it can represent a large part of posterior space. Consider three taxa, $A, B, C$ ordered alphabetically, then there are three possible topologies: $((A, B), C)$, $(A, (B, C))$ and $((A, C), B)$. The first two are represented in the cube, but the last one is missing, so at least two thirds of the posterior can be represented. In practice, posteriors will be skewed towards a dominant topology, so the first topology may have 60% support, the second 30% and leaving only 10% for the topology that is missed out. In this three taxon example, this means that 90% of the posterior can be captured. This is 30% better than using a summary tree that would only cover the first topology.

The method for learning a multivariate normal distributions over cube heights is the same as for matrices.

## Variational Bayesian approach

The application of cube space we are most interested in is Bayesian inference through a variational Bayesian distribution. To this end, we have to infer the mean vector $\mu$ and covariance matrix $\Sigma$ from an alignment and phylogenetic model. For each of the parameters of the model, we need a transformation so that the transformed parameter is normally distributed. We employ the following transformations:
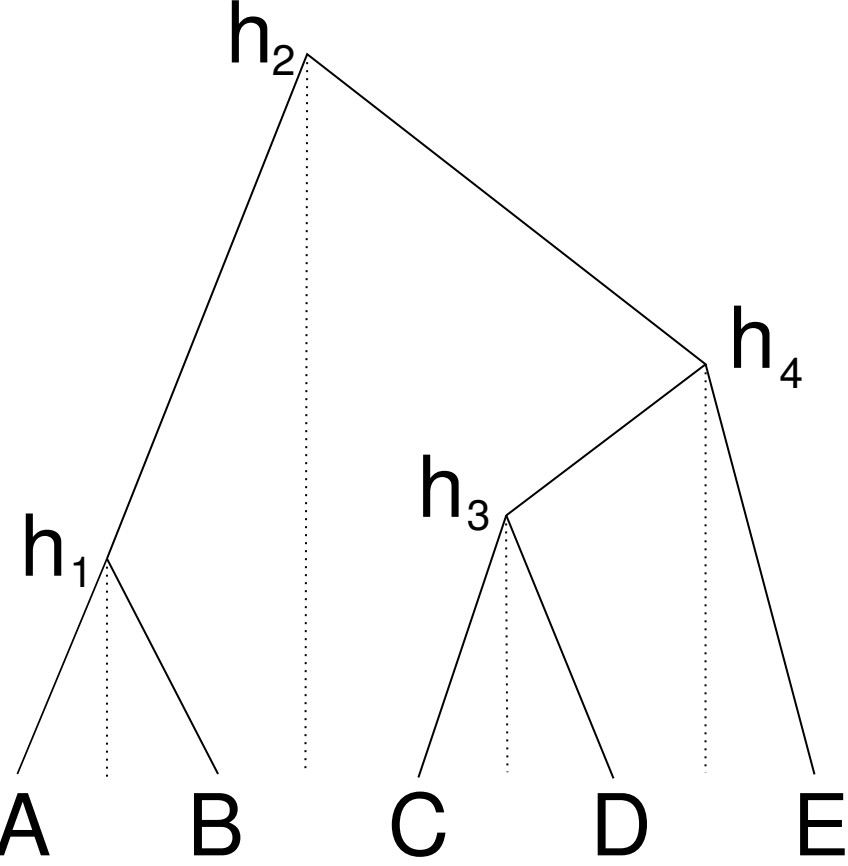

**Figure 2** Cube over five taxa consisting of a taxon ordering $A, B, C, D, E$ and heights $h_1$ for the height between $A$ and $B$, height $h_2$ for the heights between $B$ and $C$, $h_3$ for $C$ and $D$ and height $h_4$ for $D$ and $E$. Dashed lines show how the heights are associated with consecutive pairs of taxa in the taxon ordering.

- for scale parameters (like $\kappa$ of the HKY model, birth rate of the Yule model) we use a log-transform.
- for location parameters we use no transform.
- for multi dimensional parameters that are constrained to sum to 1, *e.g.* frequencies of substitution models, we use a stick breaking transform (*Betancourt, 2012*).
- for the tree we use the matrix transform. Note that internal node heights are log-transformed.

Given a phylogenetic model that specifies a tree prior, site model and branch rate model, the parameters of the model (including the tree) and associated transformations, Algorithm 3 outlines how we can infer $\mu$ and $\Sigma$. The strategy is to get a reasonably good value for $\mu$ first, and infer $\Sigma$ afterwards. To this end the first step is finding a state with high probability, ideally the maximum a posteriori probability (MAP) state, since it corresponds to the mode of the multivariate normal distribution, and thus should get us reasonably close to where $\mu$ should be.

---

**Algorithm 3** Cube-VB

---

Input: posterior distribution over a set of parameters
Input: set of parameters (including tree) and their transforms
0. Initialise tree by UPGMA
1. Find high posterior state of tree and other parameters
2. Randomly initialise ordering of taxa that is compatible with tree from step 1
3. Sample trees restricted to cube space, using MCMC
4. Estimate $\mu$ and $\Sigma$ based on transformed MCMC samples

---

The tree is initialised using Un weighted Pair Group Method with Arithmetic Mean (UPGMA) *Sokal& Michener (1958)*, a hierarchical clustering algorithm that produces a rooted time tree. The topology of the tree and timing of the tree can potentially be adjusted in the next step.

In our implementation, the tree and all other parameters are then optimised using simulated annealing (*Van Laarhoven et al., 1987*). This is a general purpose optimisation algorithm that, like MCMC, randomly explores the state space, but it differs from MCMC in the acceptance criterion of a proposed state. Higher posterior proposals are always accepted, and lower posterior proposals are accepted with a probability proportional to the change in posterior weighted by a temperature that decreases over time. We found that using a temperature schedule that decreases fairly fast to the end temperature, but repeatedly resets to the start temperature instead of a long monotonically decreasing temperature allowed escape from local minima more efficiently. For the proposals used in simulated annealing we can use standard MCMC operators implemented in BEAST 2.

To complete the cube-transform for the tree, we need a taxon ordering. Different orderings cover different parts of tree space, so finding a suitable ordering affects the suitability of the representation. This is a research question that we leave open here, but note that in the implementation using a random order compatible with the tree turned out to be sufficiently good to accurately infer parameters of interest such as tree height and tree length.

In order to estimate the covariance matrix $\Sigma$, we will generate a representative sample using MCMC. Instead of using the standard tree operators in BEAST 2, we replace the topology changing operators with a pair of operators that ensure the proposed tree can be represented by the cube restricted by the taxon ordering from Step 2 of Algorithm 3. The first operators is a variant of the narrow exchange operator (a nearest neighbour interchange operator that also proposes a new height of the node that is moved) that ensure it remains within the cube. The second operators is a subtree-slide operator (moves internal node up or down, potentially causing topology changes) and ensures that the target location is compatible with the cube ordering.

Based on the short MCMC run, we can transform the MCMC samples and directly estimate $\mu$ and $\Sigma$ from the set of transformed values. There are many tools for visualising and otherwise post-processing a posterior based on an MCMC sample, such as Tracer (*Rambaut et al., 2018*), DensiTree (*Bouckaert, 2010*), *etc*etera. Once we obtain $\mu$ and $\Sigma$, Algorithm 2 can be used to create such a sample.

# RESULTS

In this section, we validate the matrix and cube representation of tree distributions and show that the method for learning a cube based variational Bayesian distribution passes well calibrated simulation studies and we will get an impression of the algorithm's performance. Finally, we consider using a method for inferring summary trees.

## Matrices and cubes can accurately represent tree distributions

To validate whether we can accurately represent a tree posterior obtained from a full MCMC run, we construct the distance matrix for each tree by taking the logarithm of the pairwise taxon path distance, then estimate the mean and covariance matrix for all relevant entries. Next, we sample a new tree distribution using Algorithm 2 repeatedly to obtain a new tree sample. We compare clade support and clade height distribution with the original tree posterior using the CladeSetComparator from the Babel package (https://github.com/rbouckaert/Babel) in BEAST 2.

### Apes

Figure 3 shows a small illustrative example of a tree distribution over six apes. There is uncertainty at the root with 83% support for siamang being outgroup (blue trees in panels b & d), 13% orangutan being outgroup (red trees) and 4% siamang and orangutan being outgroup (green trees). A cube can only represent two out of these three cases. By representing the first two, it covers 96% of the posterior distribution, which for many practical cases may be sufficient.

Representing the third case where both siamang and orangutan are ougroup is possible with a matrix representation, as shown Fig. 3D.

What is striking in both panels a and c in Fig. 3 is that the mean of the clade heights and 95% HPD intervals are very close. However, clade support is slightly distorted: for the cube, the trees with both siamang and orangutan as outgroup is completely missing. Since this only covers 4% of the posterior, this may not be such a big issue in general. The matrix has this clade represented, but instead of the original 83%, 13%, 4% distribution has 59%, 21%, 20% distribution, heavily over representing the siamang and orangutan as outgroup. This is visible in Fig. 3C by the red dots being far from the diagonal. It appears the correlation matrix does not accurately capture possible non-linear correlations between the entries in the matrix.

One source of such non-linearities is the tree topology interfering with distance distributions between pairs of taxa: in a topology ((A,B),C), the distance between A and B and the distance between A and C can be linearly correlated. And likewise in topology ((A,C),B). But if both topologies are in the tree set there is no reason the combined correlation to always behave linearly or even approximately linearly. Use of the minimum function for clade distances in the single link clustering algorithm is another source of non-linearity. A potential remedy is to use normalising flows (*Rezende & Mohamed, 2015*) to capture such non-linearities and provide a more sophisticated transformation between a multi-variate normal distribution and the associated tree topology.

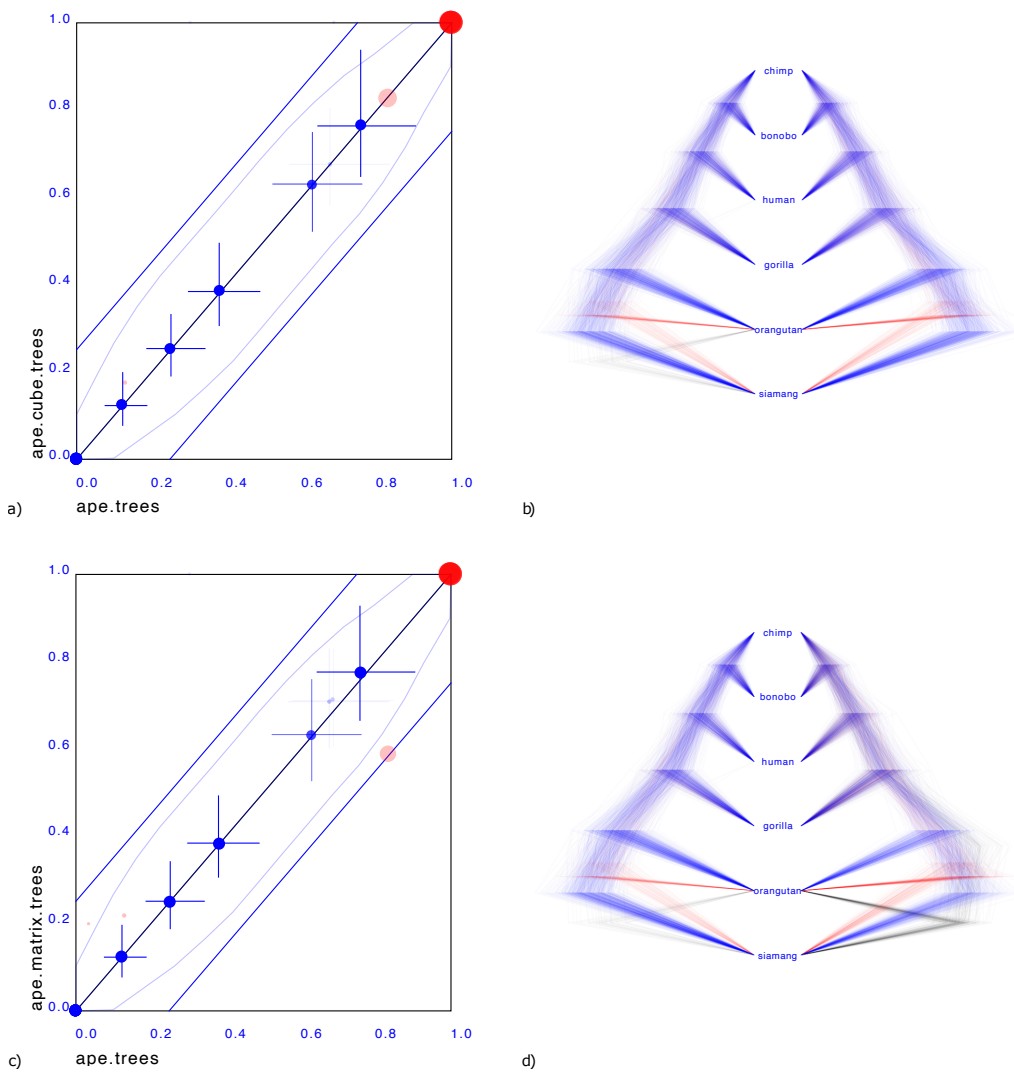

**Figure 3** **Comparing the ape dataset with a cube representation (A & B) and matrix representation (C & D).** Plates A & C show clade support in red and clade height in blue. Red dots indicate clade support, running from zero probability (bottom left corner) to 1 (top right corner). Blue dots are placed at the mean of the height of a clade. Both x and y axes are scaled so that the largest tree height (from among both data sets) is at the top right corner. Blue crosses indicate 95% highest probability density (HPD) intervals of clade height. For both red and blue dots, bigger dots mean more clade support. The x-axis represent clade values from one set, the y-axis that of another. If both sets are the same, all dots will be on the x = y line (black diagonal line). The two blue lines indicate there is 20% difference: points within these lines have less than 20% difference, points outside have more, and can be considered as evidence there is substantial difference between the two posteriors. Plates B & D show the original tree set on the left. The trees with most prevalent topology are blue, the next prevalent red and least prevalent green. Plate (B) on the left shows the cube representation and plate (D) the matrix representation. A cube accurately represents most clades, but does not cover all clades, where a matrix can represent all clades but is less accurate in clade support (see text for details).

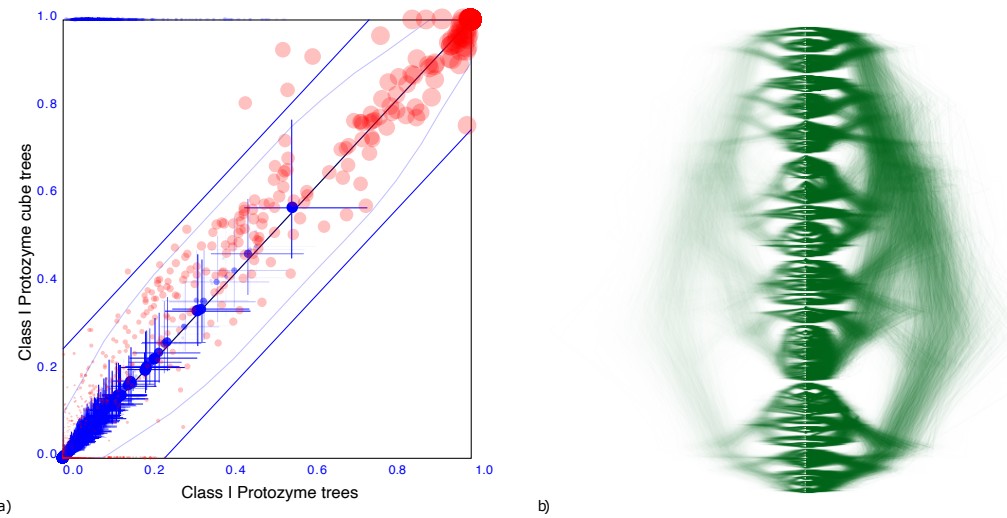

a)                                                                             b)

**Figure 4** **Comparing the AARS dataset with a cube representation.** (A) Clade support as in Fig. 3 and (B) DensiTree of original and cube tree distribution. Clade support and heights show good correspondence as does the DensiTree.

### AARS

A tree distribution with more uncertainty is from aminoacyl-tRNA synthetase (AARS) data for 204 Class I protozyme sequences (*Carter Jr et al., 2022*). A convenient way to characterise the uncertainty of a tree distribution is its entropy *Lewis et al. (2016)*, which can be calculated efficiently based on the conditional clade distribution (*Larget, 2013*) of the tree set. The AARS data has an entropy of 124.1, so more than half of the internal nodes are uncertain. The corresponding cube has an entropy of just 96.3, which shows that a cube representation can have a large entropy, thus representing a highly uncertain tree distribution.

Figure 4A shows the clade set comparison for the AARS data, which shows that most of the clades that are supported in the cube have reasonably similar support as in the original tree set. Although there is a clade with 38.9% difference in clade support, and there are 5 other clades with more than 25% clade support difference, the average clade support difference is just 4.9% for clades with over 1% support. Note the set of red dots at the x-axis, which represent the set of clades that are not represented by the cube. Furthermore, clade heights also show very close correspondences: the mean height estimate differs just 2.2%. In summary, the clades match remarkably well given the high entropy of the tree set.

Figure 4B shows a DensiTree of the original tree distribution on the left hand side and the cube representation on the right hand side. Visually, the images correspond quite well, confirming the information we have from the clade comparison panel. Note that the cube represents a large number of clades judging from the diversity of trees in the DensiTree as well as the large number of points in the clade comparison plot. Even though a considerable number of clades are missing, the majority of well supported clades are represented by the cube.
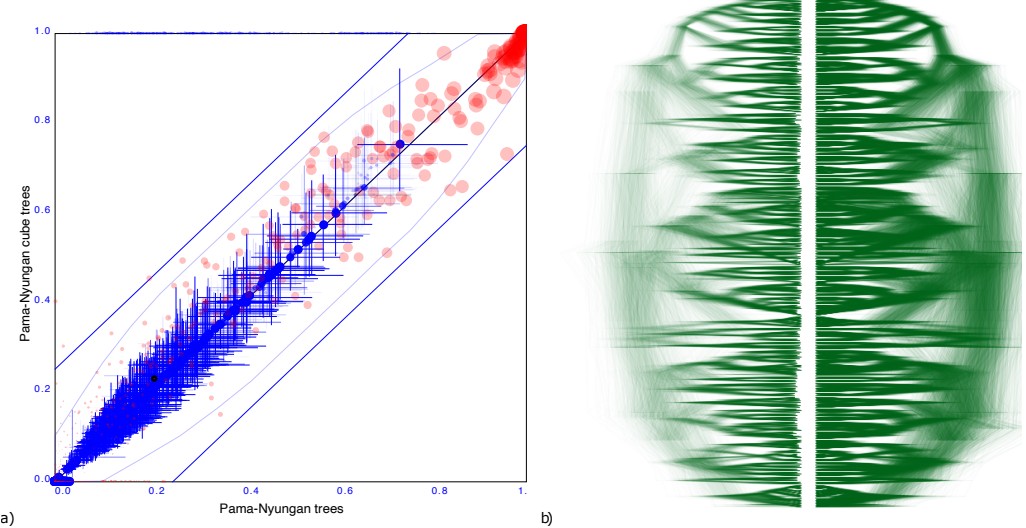

a)                                                                    b)

**Figure 5** **Comparing the Pama-Nyungan dataset with a cube representation.** (A) Clade support as in Fig. 3 and (B) DensiTree of original and cube tree distribution.

### Pama-Nynguan languages

Figure 5 shows another tree distribution, here for 306 Pama-Nynguan languages (*Bouckaert, Bowern & Atkinson, 2018*) with a large entropy of 91.7 for the original tree distribution and 78.6 for the cube. Figure 5 shows the clade comparison, showing just three clades having more than 25% clade support, and two of them are two taxon clades. Again, we see good correspondence between original distribution obtained through MCMC and its cube representation.

### Cichlids

A more troublesome example is a tree distribution over 121 Cichlid sequences sourced from *Matschiner et al. (2017)*. The entropy of the original dataset is 38.4 but that of the cube representation is higher at 49.8. Different orderings were tried based on the ordering heuristics in DensiTree, but all of them resulted in a cube representation with significant number of clades having substantially different support.

Since the cube representation contains fewer trees, there are fewer terms to contribute to the entropy, and since all terms must add to 1, these terms can be expected to be larger than for the original distribution. Therefore, one would expect entropy of a cube representation to decrease from that of the original tree distribution The fact the entropy for the cichlids data increased in the cube representation compared to the original dataset can be interpreted as a hint that the ordering is less than optimal. Optimising orderings is a topic for further research, but this example shows it is crucial for a good tree set representation.

Figure 6 shows the clade comparison and DensiTree. There are 23 clades that have more than 25% difference in support and the maximum difference in clade support is 80% for a clade in the original set that is missing from the cube representation. The mean height

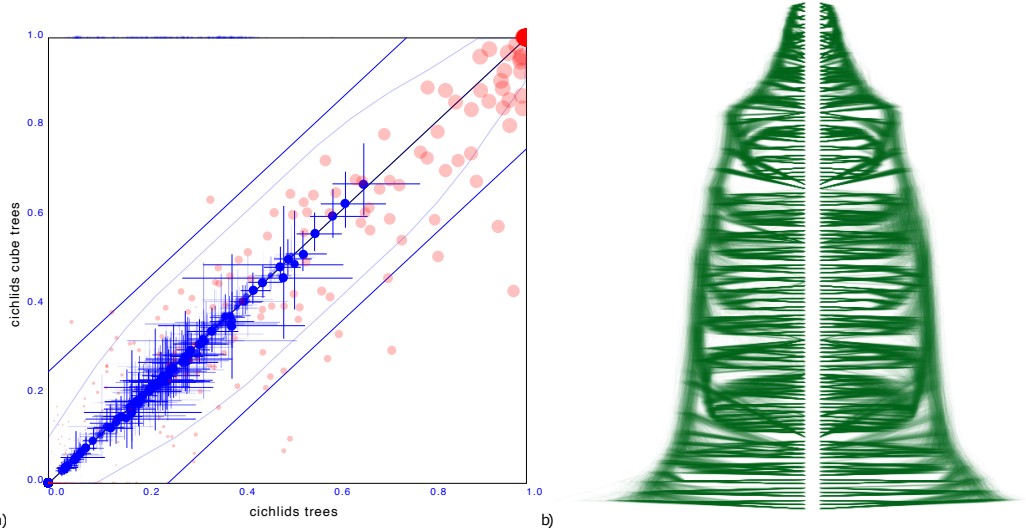

**Figure 6** **Comparing the cichlids dataset with a cube representation.** (A) Clade support as in Fig. 3 and (B) DensiTree of original and cube tree distribution. Significant number of clades have large difference in support in the cube representation, though heights of those clades that are captured show good correspondence.

of clades differ only 0.57%, so correspond quite well. In this case, a cube representation appears to be less satisfactory than in the other examples.

### Indo-European languages

To demonstrate the cube representation works well for internal nodes, even though tip dates are sampled, we considered a recently published tree distribution over 161 languages (*Heggarty et al., 2023*). The tree set has an entropy of 42.3, the cube has 36.6 (Figure 7).

Maximum difference in clade support is 28.5% but only 2 clades have difference of 25% and on average the clade support difference is 4.1% for clades with more than 1% support. Mean height difference of clades (not counting tip heights) is less than 0.79% on average. So, even though the original distribution has dated tips, and the cube does not, clade support and clade heights remain quite closely represented when compared to the original.

### Cube-VB algorithm passes well calibrated simulation study

To verify that the cube based variational distribution algorithm works, we performed a well calibrated simulation study (*Mendes et al., 2023*; *Talts et al., 2018*) using a tree with 50 taxa sampled from the Yule tree prior with birth rate narrowly distributed as normal (mean = 6, sigma = 0.1). This results in tree heights with mean 0.58 and 95% HPD of 0.34 to 0.79. Sequence data is generated under a HKY model with kappa lognormally distributed (mean = 1, sigma = 1.25), and a strict clock with rate fixed at 1 is used to complete the model. So, the parameter space being sampled consists of the kappa and birth rate parameters and a tree. We use sequence lengths of 100, 250 and 500 resulting in observed entropy in the inferred tree posteriors of 10.6, 6.4 and 3.8 respectively (Table 1). We run 100 instances
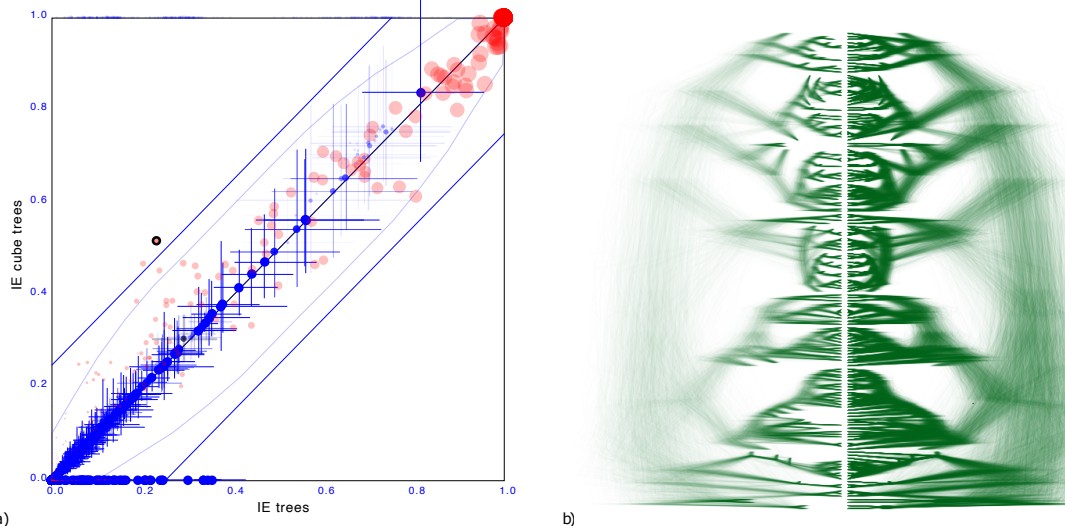

a)    b)

**Figure 7** **Comparing the Indo-European dataset with a cube representation.** (A) Clade support as in Fig. 3 and (B) DensiTree of original and cube tree distribution. Even with dated tips the cube representation remains accurate.

**Table 1** **Coverage (in %) of 100 runs over 50 taxa with Yule, HKY strict clock for three different sequence lengths.** Longer sequences result in lower entropy, but coverage of the parameters of interest (last four lines in table) are all in acceptable range.

| Sequence length: | 100 | 250 | 500 |
|---|---|---|---|
| Mean and 95% HPD Entropy | 10.6 (5.9–14.6) | 6.4 (3.2–10.4) | 3.8 (1.3–6.3) |
| Tree height | 95 | 94 | 93 |
| Tree treeLength | 94 | 95 | 96 |
| BirthRate | 92 | 94 | 94 |
| Kappa | 94 | 90 | 93 |

for each sequence length, and expect the 95%HPD of parameters of interest to contain the true value from 91 to 99 of the cases for the test to pass. Table 1 shows the results and the simulation study indeed passes.

A slightly more challenging case is where we sample trees from a coalescent tree prior with constant population size sampled from a lognormal (mean = 0.1, sigma = 0.1). The tree heights have a mean of 2.3 with 95% HPD interval of 0.6 to 5.0. We use a site model with gamma rate heterogeneity where the shape parameter is sampled from an exponential with mean 1, and use a GTR model with rate CT fixed to 1 and rate AG sampled from a log normal with mean 1 in real space and sigma = 1. The other four transversions rates are sampled from a log normal with mean in real space of 0.5 and sigma = 1. Stationary frequencies for the GTR model were sampled from a Dirichlet with $\alpha = 4$ for all four dimensions. For this setup, getting accurate estimates of the uncertainty required running the MCMC a bit longer than for the case using HKY. Table 2 shows the coverage we get

**Table 2** As **Table 1** but for coalescent with constant population, GTR with estimated frequencies using gamma rate heterogeneity and strict clock for three different sequence lengths. Coverage of the parameters of interest (last thirteen lines in table) are all in acceptable range.

| Sequence length: | 100 | 250 | 500 |
|---|---|---|---|
| Mean and 95% HPD | 13.1 (5.9–15.7) | 5.7 (3.0–9.1) | 4.7 (1.5–6.5) |
| Entropy | | | |
| Tree.height | 91 | 90 | 97 |
| Tree.treeLength | 90 | 91 | 97 |
| rateAC | 92 | 95 | 98 |
| rateAG | 100 | 98 | 95 |
| rateAT | 92 | 95 | 94 |
| rateCG | 93 | 96 | 94 |
| rateGT | 92 | 99 | 95 |
| freqParameter.1 | 94 | 92 | 96 |
| freqParameter.2 | 96 | 92 | 96 |
| freqParameter.3 | 95 | 95 | 97 |
| freqParameter.4 | 95 | 96 | 98 |
| gammaShape | 93 | 100 | 97 |
| popSize | 90 | 90 | 90 |

[1] If the 95% HPD estimates are correct estimates, the simulation study is an experiment of flipping a coin with 95% probability of turning heads repeated 100 times. Here, observing a head equates finding the true value of a parameter being inside the 95% HPD interval. This results in a binomial distribution of observing heads on average 95 times and a 95% probability of being within a range of 91 to 99. Observing almost all coverage values in this range with the occasional outlier at 90 or 100 is deemed acceptable.

for all paramaters of interest and they are all in an acceptable range,[1] although population size coverages were on the low side.

This demonstrates that the cube based variational distribution algorithm can work well estimating a tree, scale parameters as well as constrained sum parameters (*i.e.* frequencies) for a number of popular phylogenetic models. Many model implementations are available in BEAST 2 that only require these types of parameters, in particular nucleotide and amino acid substitution models.

## Cube based variational distribution algorithm performs well

To see how well the cube based variational distribution algorithm scales with number of taxa, we ran a simulation study with 50, 100, 250, 500, 750 and 1,000 taxa. For each taxon set size, we ran 25 instances of the Yule/HKY/strict clock model used for the well calibrated simulation study. For each instance, alignments of 250 sites were generated. We checked that coverage of 95% HPD intervals for tree height, tree length, kappa and birth rate parameters was between 22 and 25 to verify that the inferred posterior indeed matches the true distribution. Experiments were performed on a 2021 MacBook Pro laptop in OS X using BEAST v2.7.5 and BEAGLE v4.0.0 (*Suchard & Rambaut, 2009*).

Table 3 shows run times in seconds, demonstrating it is feasible to perform large analyses in a short span of time. Sampling a representative posterior trace and tree log took the least time taking up no more than 2% of total runtime. MCMC for accurately estimating the covariance matrix took most of the time, from 40% for smaller taxon sets to over 90% for the larger tree sets. MAP finding and Cholesky decomposition in comparison took less time and are taking up relatively less time with growing taxon set sizes.

**Table 3** **Run times in seconds averaged over 25 instances for various taxon set sizes (in columns).** Numbers in brackets are standard deviations.

| Number of taxa | 50 | 100 | 250 | 500 | 750 | 1000 |
|---|---|---|---|---|---|---|
| MAP finding | 1.17 (0.03) | 1.48 (0.03) | 2.36 (0.08) | 3.91 (0.08) | 5.30 (0.19) | 6.43 (0.40) |
| MCMC | 1.15 (0.03) | 2.77 (0.05) | 12.08 (0.24) | 41.84 (0.72) | 94.74 (3.29) | 172.25 (8.24) |
| Decomposition | 0.45 (0.01) | 0.60 (0.02) | 0.93 (0.03) | 1.69 (0.04) | 2.88 (0.20) | 4.47 (0.10) |
| Sampling output | 0.23 (0.02) | 0.36 (0.02) | 0.68 (0.02) | 1.40 (0.03) | 2.18 (0.05) | 3.27 (0.15) |
| Total runtime | 3.00 (0.05) | 5.20 (0.08) | 16.05 (0.27) | 48.83 (0.79) | 105.10 (3.33) | 186.42 (8.60) |

Total calculation time fits a quadratic function ($1.70E - 4x^2 + 1.43E - 2x$) with $R^2 = 0.9997$, so the algorithm seems to scale quadratically. The chain length used are a linear function of the number of taxa, and tree likelihood calculation time (which dominates posterior calculation time here) both scale linear in the number of taxa. Changing the former could change how the algorithm scales. The most striking feature of the table though is that all times can comfortably be expressed in seconds using just three digits before the decimal point.

### Summary tree

Observe that by setting $z = 0$ in Algorithm 2 we can obtain a tree that is at the mode of the distribution. This tree should be representative of the complete distribution so can be used as a summary tree. Note that it can have a topology that differs from all trees in the tree set to be summarised, unlike popular methods like the maximum clade credibility (MCC) tree implemented in TreeAnnotator. Table 4 shows some the sum of posterior clade support in the summary tree for MMC and matrix representation based summary trees. This is proportional to the number of internal nodes minus the Robinson-Foulds distance averaged over the tree set. The table shows that matrix based summary tree never produced worse trees and sometimes does considerably better. The log product of clade probability is used in MCC trees to find the best fitting topology. Table 4 shows even according to that criterion the matrix approach can do better, which is remarkable since MCC explicitly maximises that criterion. Further evaluation with less crude measures of summary tree quality are required to establish when matrix based summary trees outperform MCC trees.

## DISCUSSION

**Matrix and cube representations can accurately represent tree distributions:** For tree distributions with low to medium entropy (*Lewis et al., 2016*) based on its conditional clade distribution (*Larget, 2013*), we demonstrated that a substantial part of the posterior can be represented using a single cube. For those cases it is possible to do fast inference of rooted time trees using a cube based variational distribution approach outlined in Algorithm 3. For tree distributions with high entropy, it may not be possible to capture a substantial portion of the posterior by a single cube. An example is a sample over 50 taxa from a Yule prior (Fig. S1), where a cube representation will suggest much older internal node heights with higher clade support than the original distribution. Using more

**Table 4** **Sum and log product of posterior clade support over all clades in the summary tree for MCC tree and matrix representation based tree.** Higher is better. The entries marked with asterisk ignore zero support clades (one clade for cichlids, three for NZ votes). The matrix approach is never worse and sometimes considerably better.

| Data set | Sum of clade probability | | | Log product of clade probability | | |
|---|---|---|---|---|---|---|
| | MCC | | Matrix | MCC | | Matrix |
| Apes | 4.8 | = | 4.8 | −0.2 | = | −0.2 |
| IE | 136.9 | < | 138.9 | −35.0 | < | −29.2 |
| AARS | 133.6 | < | 142.9 | −128.4 | < | −90.8 |
| Cichlids | 91.7 | < | 93.7 | −42.3 | < | −34.7* |
| NZ votes | 26.7 | < | 34.1 | −297.1 | < | −217.6* |
| Pama-Nyungan | 255.6 | < | 262.4 | −80.5 | < | −59.3 |
| Species | 27.5 | = | 27.5 | −2.6 | = | −2.6 |

than one cube might elevate this somewhat, but since there can be overlap between the trees represented by the various cubes, and this overlap is hard to characterise (L. Berling, 2023, personal communication) this may not be a fruitful avenue. A more productive way may be to use more entries in the distance matrix. This breaks the bijective property of the transformation, but facilitates a greater coverage of the full tree space. This will result increase the non-linearity of dependencies between the various matrix entries due to correlations changing when topologies change. Normalising flows (*Rezende & Mohamed, 2015*) allow for greater flexibility in representing such non-linear dependencies, and may provide a way to transform a multi-variate normal distribution into a tree distribution.

One situation where node height distributions can deviate significantly from being log-normally distributed is when node calibrations are used in the tree that interact with each other. For example, clade $A$ may be known to be at least $X$ years old, so a uniform prior with lower bound of $X$ seems appropriate. When clade $B$ (consisting of clade $A$ plus 1 node so that the MRCA of $B$ is the parent of the MRCA of $A$) is assigned a distribution with a mean not very far above $X$ (with large variance), the age distribution of the MRCA of $B$ will be lower bounded by $X$ as well. In this case, a log-transform may not be appropriate for accurate capturing of this distribution, and a matrix representation may fail to accurately capture such distribution. Again, normalising flows could be employed to address this.

**Optimisation based variational Bayesian:** Variational Bayesian algorithms are usually optimisation based, where the evidence lower bound (ELBO) is maximised. The ELBO is proportional to the posterior plus a term that measures the distance between the true and approximate distribution. The benefit of maximisation over MCMC is that there are no restrictions on optimisation proposals unlike with MCMC where one needs to be careful to use appropriate Hastings ratios in order to ensure there is no bias in sampling from the posterior. One way to optimise the ELBO is by the hugely popular stochastic gradient ascent algorithm, which randomly draws a new state, calculates the gradient of the ELBO with respect to a random subset of parameters and moves parameters in the state in the direction of the gradient. This process is iterated until convergence is reached. However, there are some issues with gradient ascent: a gradient is required, which can be computationally
expensive. Furthermore, it has domain specific parameters, such as learning rate, learning rate scheme, and parameter subset size. Also, random subsets may need to be tuned for efficient exploitation of subsets. Finally, the convergence criterion has large impact on the number of iterations. Since points are drawn stochastically, the convergence criterion needs to deal with noise. Instead of dealing with all these issues, here we introduced an MCMC based algorithm to learn a variational distribution that benefits from all MCMC related optimisations implemented in BEAST 2 (such as adaptive operator tuning). It serves as a proof of concept demonstrating that cube spaces are suitable for variational inference. Of course, this still leave opportunities for ELBO maximisation implementations of cube based variational inference.

**Cube-VB algorithm parameters:** Notwithstanding the issues of ELBO optimisation based VB, the MCMC based approach to learning a variational distribution we introduced in Algorithm 3 has a number of parameters that need to be specified, in particular the start and end temperatures and number of repeats for simulated annealing for finding a (close to) MAP state. We chose a start temperature of 0.1 and end temperature of 0.01, which worked quite well for our experiments, so this seems a reasonable option. Putting in more than 25 repeats barely led to improvement of the posterior of the MAP state, but halving it turned out to be detrimental (well calibrated simulation studies did not pass).

Another parameter to set is the length of MCMC chain for estimating the covariance matrix. The heuristic implemented in the cubevb package for BEAST 2 is to count the number of parameters and multiply with a user specified value. In our experiments with the HKY model, a multiplier of 1,000 was sufficient. The more samples, the more accurate the matrix estimate but the longer it takes for the algorithm to finish. In that sense, it nicely fits the class of online algorithms (*Bouckaert, Collienne & Gavryushkin, 2022*) that at any time can provide an answer, but the answer becomes more accurate the longer the algorithm runs. Detecting when longer runs do not increase accuracy of the posterior any more is an open question where automated convergence techniques employed for standard MCMC (*Berling, Bouckaert & Gavryushkin, 2023*) could be exploited. These are all based on running two analyses in parallel and measuring the difference in the distributions that are obtained.

**Why the Cube-VB algorithm works:** Since plain MCMC is used in Algorithm 3 one may wonder why this is more efficient than just running full MCMC and not being restricted to a cube space. The reason for this stems from the assumption that the true distribution can be represented as a transformation of a multivariate normal distribution. Consider the $\kappa$ parameter for the HKY model and a log-transformation. In practice, we see that $\kappa$ is close to log-normally distributed, so representing the posterior as a normal with log-transform leads to little distortion. Now notice that we only have to estimate the mean and variance of the log-normal under this assumption, instead of estimating a full unparameterised distribution, which is what we would be doing with full MCMC. So, even though a small MCMC sample will not be sufficient to accurately represent the distribution of $\kappa$ and statistics like 95% HPDs, there will be sufficient information to accurately estimate two parameters of a log-normal, from which statistics like 95% HPDs then can accurately be inferred.

**Variational Bayesian for matrix representation:** The Cube-VB algorithm can be adapted to suitable matrix representations by replacing Step 2 by some method of identifying which entries in the matrix need to be specified and which entries can be left unspecified. Step 3 would need adjustment in the operators to ensure all possible tree topologies that can be represented by the matrix are available to be proposed. It is an open problem what the best strategy is for identifying which and how many entries should be specified. Since a cube can capture a substantial part of a posterior, identifying a cube could be a first step followed by filling in entries where clades represented have substantial support.

## CONCLUSION

Matrix space is introduced as a representation of all or, in the case of cubes, a significant part of tree space. We demonstrated for simple phylogenetic models it allows fast inference of the most relevant part of the tree posterior using just a cube representation, thus capturing most of the uncertainty in the posterior distribution.

There are many venues to extend and develop this work, in particular into the area of more sophisticated phylogenetic models. Here we mention just a few of the more commonly used models and potential issues implementing inference for them in matrix representation, in particular cubes:

- In general, finding an appropriate subset of pairwise distances, *i.e.* subset of entries in the matrix, an obvious open problem. Finding the optimal ordering for a cube is an instance of this problem. It is probably related to finding a good ordering for drawing trees in DensiTree (*Bouckaert, 2010*) for which various heuristics have been developed. It is not clear how to transfer these heuristics to the VB setting and new heuristics may need to be developed.
- Flexible nonparametric tree priors take epochs in accounts like the Bayesian skyline plot (*Drummond et al., 2005*) and birth death skyline plot (*Stadler et al., 2013*) and are widely used but may be challenging to parameterise in a variational Bayesian setting due to their discrete components. Models that integrate out most parameters and have only a single continuous parameter, like BICEPS and multi Yule (*Bouckaert, 2022*), can be good alternatives and should be straightforward to implement in our cube VB algorithm.
- Relaxed clocks (*Drummond et al., 2006*; *Douglas, Zhang & Bouckaert, 2021*) need to be adapted for cube space. Due to discontinuities between different topologies, simply associating a (log)normal branch rate distribution with each taxon and each dimension of the cube may not be sufficient. Normalising flows (*Rezende & Mohamed, 2015*) may be required to capture relaxed branch rate distributions.
- Dated tips put restrictions on the height of internal nodes. When tip dates are uncertain and need te be sampled from a distribution, this adds extra dimensions to the multivariate normal representation of the posterior.

On the other hand, extensions into empirical models of amino acids, models that integrate out internal states like ancestral reconstruction (*Lemey et al., 2009*) and phylogeography (*Bouckaert, 2016*) should be straightforward to adapt.

In summary, there are many exciting new opportunities waiting to be explored in cube and matrix space for fast phylogenetic inference.

## ACKNOWLEDGEMENTS

This work was greatly helped by stimulating discussions with members of the Center of Computational Evolution at the University of Auckland, in particular with Alexei Drummond. Special thanks to Jordan Douglas for proofreading.

### Funding

The authors received no funding for this work.

### Competing Interests

The authors declare there are no competing interests.

### Author Contributions

- Remco R. Bouckaert conceived and designed the experiments, performed the experiments, analyzed the data, prepared figures and/or tables, authored or reviewed drafts of the article, and approved the final draft.

### Data Availability

The data and code is available at GitHub and Zenodo:

- https://github.com/rbouckaert/cubevb

- Remco, B. (2024). CubeVB package for BEAST2: archive of https://github.com/rbouckaert/cubevb. In Variational Bayesian Phylogenies through Matrix Representation of Tree Space: Vol. tbd (v1.0.1). Zenodo. https://doi.org/10.5281/zenodo.10594658.

### Supplemental Information

Supplemental information for this article can be found online at http://dx.doi.org/10.7717/peerj.17276#supplemental-information.

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
