# Peer review of "Variational Bayesian phylogenies through matrix representation of tree space"

_PeerJ, doi:10.7717/peerj.17276_

## Round 0.1 · original submission · Minor Revisions

Two reviewers have commented on your manuscript - recognising both the rigor of the presented work and its relevance to others in the field of phylodynamic modelling and inference.

Reviewer 1 poses a number of specific questions that would help clarify the paper for a general reader, and some methodological questions regarding improved search algorithms, and appropriateness of the model for imbalanced and high-entropy systems.

Reviewer 2 questions the 'Variational Bayes' epithet for the methodology described and requests a number of clarifications with respect to related work, the reliance on UPGMA tree ordering and applicability for time based phylodynamics.

Reviewer 2 also points out that the general way the relationship between time trees and matrix spaces is introduced tends to obscure the core focus on cube space - suggesting the theoretical treatment might be appendicised. They also make some suggestions to further illustrate the conceptual relationships that could be helpful to the reader.

Both note a number of typos, and Reviewer 2 further requests the accompanying code and data is checked for problems and (at least) additional documentation provided to support reproducibility.

Please also find my own comments below:

Overall I found the manuscript clear and reasonably well focused. I do however agree with reviewer 2 that whilst the (quite clear) introduction sets the reader up for a discussion of alternate tree space representations, there is a subtle disconnect between the methods and results section which will certainly confuse readers less knowledgeable about the field (which perhaps includes myself). I also felt that the original motivating reason for this work - scalable inference - was not convincingly addressed. A revised presentation of the relative computational cost of the cube approach for increasing system size might help, as well as a clear summary statement in the concluding remarks.

Embedding the introduction and methods more fully in the literature, and explaining some of the more precise mathematical language in practical terms will help. I also offer a number of specific points of revision:

R1. Normalising flow (Rezende and Mohamed 2015) is mentioned twice in the context of better capturing nonlinearity in the input matrix. It would help to briefly describe how and why (biologically) such non-linearities arise in the introduction & methodology sections, and provide perhaps one additional sentence in the dscussion to summarise what 'normalising flow' actually accomplishes in this context.

R2. Clearer legend and use of colour in Figure 3 - the first figure to show clade support plots and densitrees.

i. The use of red and blue both in the plots and the densitree figures is confusing. Recommend use of a different colour or visual mark for clade support, or alternatively exclude red and blue from densitree's palette (as seems to be the case in later figures).

ii. The legend for Figure 3 should identify all marks in panels a and c - 2D scatter plots showing relationship between support for different tree representations: please include in the legend what what the blue lines parallel to the diagonal and the curved pale blue lines mean. The legend should also indicate what values are encoded by the diameter of circles on the plot, and (ideally) what the error bars capture.

R3. What does 'all in acceptable range' mean to a statistics-literate layreader in the context of the coverage percentages reported for the 13 parameters in table 2 ?

R4. Grammar / Typos
i. Figure 6 and 7 in the assembled PDF have the same graphic (though clearly should be and are different in the individual artwork) - please check.

ii. Line 365-366 - this sentence appears incomplete.

iii. Line 383 - ..'implemented in out cube'.. should be 'our'

iv. Line 394-395 - reword - unless of course "exciting opportunities" really are waiting to explore cube and matrix space !

Reviewer 1 ·

Basic reporting

No comment.

Experimental design

Here I have a few suggestions concerning the experiments that the author performed:

1. Could the author provide an explicit comparison of the computational cost between a full MCMC algorithm and the CubeVB method?

2. Could the author provide some possible explanation why CubeVB does not perform well for the cichlids dataset?

3. Could the author include datasets with larger observed entropy as well?

Validity of the findings

No comment.

Additional comments

The author has developed a variational Bayesian algorithm for inferring rooted trees based on matrix representation, which is a significant step towards scalable phylogenomic inference methods. Here I have a few additional comments:

1. The authored has used simulated annealing to escape local minima in MCMC. Can parallel tempering perform better?

2. The author has used multivariate normal to approximate the distribution of the transformed parameters. Shall we consider the case when the true distribution is quite skewed?

3. When factors like gene flow exist, could the cube representation still capture the primary part of the tree space?
 
4. Please check typos in the text. For example, in Algorithm 1, "While There is more then"; P5, L166, "get and impression"; P11, L263, "they ar all", etc.

·

Basic reporting

This paper shows that the following procedure can provide a useful intermediate between a fixed tree analysis and a full Bayesian analysis:

* start by building a UPGMA tree
* perform MCMC on trees constrained to have the same taxon ordering
* summarize these trees in terms of their coalescence times using neighboring clades
* fit a multivariate normal to these coalescence times and other (transformed) parameters

The paper is novel and thought-provoking. This seems like it could be a useful approximate approach and the validations are interesting. The analysis of the results is appropriate and the figures are fairly clear. The exposition could be substantially improved in several ways, as described below, and in places obscures understanding. For example, the manuscript has many typos, such as "Euclidian", "urangatan", "correspondance", "and get and impression", etc. There are many tools that can help with such problems, such as latex spell checkers, the LTeX extension for VS Code, which does LanguageTool grammar/spell checking, not to mention ChatGPT. Please have a colleague read over the paper before resubmission.

Importantly, this is not variational Bayes according to any definition I've seen. Variational Bayes is done through optimization. Furthermore, it would seem difficult to optimize an objective that is based on the approach in this paper, which isn't differentiable when two node heights pass one another. These issues go beyond the statement in the paper: "However, there are some issues with gradient ascent: a gradient is required, which can be computationally expensive."

Rather than being about variational Bayes, this paper feels a lot more in spirit to the Hoehna/Drummond, Larget, and Zhang/Matsen papers on representations of tree space. In all of these papers, like the one here, one can use a short MCMC run and then fit some parameterized distributions on trees to it. Again, none of these papers called themselves "variational Bayes"-- this term came later in some subsequent work.

Experimental design

However, something that all of these papers did is explore the extent to which the parameterized distribution improved the posterior estimates. For example, the Larget paper showed that CCD improves probability estimates for less-likely trees. In this paper, there is a description of why it might work in "Why MCMC based variational Bayesian works", but as far as I can tell the empirical comparisons done are between full MCMC run and the cube method, without comparing the short MCMC run for the cube method to the cube method.

One of the primary use cases for BEAST is in phylodynamics, in which one typically has samples through time. Does the cube algorithm work for this setting? If not this should be made clear.

In general, the reliance on UPGMA feels like a weakness. Before I used the cube approach, I'd want a broader-scale exploration of if UPGMA returns an appropriate taxon order.

Because UPGMA returns a tree and not a planar tree, there is some flexibility in terms of taxon ordering. Some of these should be better than others for the cube space. Namely, orderings that bring clades together that may themselves mingle to form good clades are better than orderings that don't. It appears that the current clade ordering uses a random planar embedding of the tree, which seems suboptimal.

Validity of the findings

The findings appear valid.

Regarding reproducibility, there is a GitHub repo with links to ingredients used in the paper. However, it would be a lot nicer to have a script/notebook/something that could be used to make the plots in the paper, or at least a README in the .tgz file describing the data. Note that I got errors like this when trying to unpack wcss.tgz:
wcss/yule-hky-strict50vb/input50vb.xml: Skipping hardlink pointing to itself: wcss/yule-hky-strict50vb/input50vb.xml

Additional comments

The research focus is really on the cube space, although the description really starts first with a more general setting, and then specializes on the cube space. My suggestion is to focus directly on the cube space, with perhaps the more general setting as an appendix. In any case it seems that there is a bijection between the cube space and the set of rooted time trees that can be expressed with a given taxon ordering. A precise statement would be helpful here.

It would seem helpful to connect the presentation here with prior literature. The representation in https://doi.org/10.1038/s41467-022-31511-0 is certainly related, though I'd imagine that the literature on representing trees by coalescence times goes well before that.

This certainly isn't a requirement, but I think it would be interesting to have a figure showing the various topologies that are represented by a disk in cube space of a reasonable radius and shape for a few mean locations. This would give a flavor of what a normal distribution would give you.

There appear to be some problems with Algorithm 3. There are two input lines-- is that on purpose? Is step 2 supposed to refer to step 0?

Although the runtime does seem fast, Table 3 is difficult to interpret without some sort of baseline for comparison.


Details:

Abstract:
First sentence of abstract: what is the justification for this? That seems to be the claim of the manuscript.
Should be "single linkage clustering".
Suggest "package for BEAST 2".

L28: should be "Variational Bayesian"
L29: VB is not restricted to multivariate normal, though this is the most common case
L39: See https://arxiv.org/abs/2204.07747 for an extension to time trees
L59: should be "single linkage"
L62: "is a finite" should be "is finite"

Fig. 3:
* could we have some cleaner axes labels for the panels a and c?
* I'm confused that 3a shows the red dots on the diagonal, which if I understand correctly means that the cube support is correct, whereas the more flexible matrix representation, which includes the alternate outgroup, is not.

L113 "In practice, posteriors will be skewed towards a dominant topology, so the first topology may have 60% support, the second 30% and leaving only 10% for the topology that is missed out. In this three taxon example, this means that 90% of the posterior can be captured." Is this a general statement?

L203 "Based on the conditional clade distribution (Larget, 2013), it has an entropy of 124.1, so more than half of the internal nodes are uncertain.": "entropy" doesn't appear in this paper-- perhaps you should refer to the Lewis paper here? Also, it seems like one can directly compute the fraction of uncertain nodes-- why do we need CCD?

L214 "Fig.4b shows a DensiTree of the original tree distribution on the left hand side and the cube reprentation on the right hand side. Visually, the images correspond quite well, confirming the information we have from the clade comparison panel.": I feel like this is a bit of an unfair comparison because the DensiTree uses a compatible taxon ordering with the cube algorithm.

---

## Round 0.2 · Minor Revisions

Thank you for carefully considering all points raised in the first round of revision, particularly the various clarifications and expansions which I am certain will help readers not so familiar with the domain to understand the ramifications of your work.

I've noted in the attached PDF a number of very minor typos that should be addressed before online publication. Of these, two are relatively more complex:

1. Figure 2's depiction of the cube representation shows the inter-leaf mrca distances on a slanted cladogram - which may for some indicate the cube distances are projection dependent. Should this figure be redrawn to more accurately indicate the distance ?

2. Line 405 - the last sentence in the discussion appears to not be finished ?

Feel free to contact me if you need anything clarified. Looking forward to seing this out in PeerJ!

·

Basic reporting

no comment

Experimental design

no comment

Validity of the findings

no comment

Additional comments

Congratulations on a nice revision. This paper is ready to ship.

I do think that adding the statistics of the short run would strengthen your argument. One can take an insufficient run, do a transformation, and then get a good run.

---

## Round 0.3 · accepted · Accept

Thanks for addressing these final issues - once again, I look forward to seeing how this approach is taken up by the community.